# Factors associated with miscarriage in Nepal: Evidence from Nepal Demographic and Health Surveys, 2001–2016

Sharadha Hamal[1]*, Yogendra B. Gurung[2], Bidhya Shrestha[2], Prabin Shrestha[3], Nanda Lal Sapkota[4], Vijaya Laxmi Shrestha[5]

1 Gandaki Medical College, Tribhuvan University, Kirtipur, Pokhara, 2 Central Department of Population Studies, Tribhuvan University, Kirtipur, Kathmandu, Nepal, 3 Tri-Chandra Multiple Campus, Tribhuvan University, Kirtipur, Kathmandu, Nepal, 4 National Statistics Office, Thapathali, Kathmandu, Nepal, 5 National Academy for Medical Sciences, Kathmandu, Nepal

* dipshayu@gmail.com

**Data Availability Statement:** "Data may be obtained from DHS (https://dhsprogram.com/data). To obtain data from the DHS website, a data requisition application is required. The authors

## Abstract

### Background

Miscarriage is a major public health concern in low and middle-income countries (LMICs) like Nepal. This study aims to examine the factors associated with miscarriage among pregnant women of reproductive age (15–49 years) in the past 15 years.

### Methods

There were a total of weighted sample of 26,376 cross-sectional pregnancy data from Nepal Demographic and Health Surveys (NDHS) 2001, 2006, 2011, and 2016 combined together, which was used in the study. Multilevel logistic regression analysis that adjusted for cluster and survey weights was used to identify factors associated with miscarriage among pregnant women of reproductive age in Nepal.

### Results

The results showed that maternal age, contraception, tobacco smoking, wealth index, respondents' educational status, and, caste/ethnicity were found to be strong factors of miscarriage in Nepal. The likelihood of having a miscarriage among older women (≥40 years) was more than 100% (aOR = 2.12, 95% CI [1.73, 2.59]), among non-users of contraception was 88.9% (aOR = 1.88, 95% CI [1.68, 2.11]) (p<005) and non-smoking women had a 19% lower odds of miscarriage (aOR = 0.81, 95% CI [0.69, 0.95]). Respondents from the richest wealth index had 50% (aOR = 1.50, 95% CI [1.22, 1.85]) higher likelihood of miscarriage. Mothers with only primary education had a 25% higher chance of miscarriage (aOR = 1.25, 95% CI [1.09, 1.44]) compared to those with secondary and higher secondary education. In relation to caste/ethnicity, Dalits had 13% lesser likelihood (aOR = 0.87, 95% CI [0.74, 1.02]) and Janajatis had 26% lower chances of a miscarriage than Brahmin/Chettri (aOR = 0.74, 95% CI [0.64, 0.85]).

confirm that others may acquire the data in the same way the authors did and that the authors did not receive any special consideration from MEASURE DHS/ICF International. The titles of the data set used for this study were: NPKR81FL, NPIR81FL, and NPGR81FL."

**Funding:** The author(s) received no specific funding for this work.

**Competing interests:** The authors have declared that no competing interests exist.

## Conclusion

Findings from this study show that miscarriages are associated with maternal age, use of contraception, smoking, wealth index, caste, and ethnicity. Interventions aimed to improve use of contraceptives, avoiding smoking and pregnancy planning on the basis of maternal age, are needed to prevent miscarriage. Also, women from Brahmin ethinicity and those with the highest income index require greater attention when it comes to miscarriage prevention strategies in Nepal.

## Introduction

Miscarriages are the most common public health risk that could occur in every pregnancy [1]. Women who undergo miscarriages continue to suffer in secret, endure painful events, and face societal stigma, which causes them to experience emotional loneliness and mental health issues. It is a terrible experience, especially in nations with low to middle income like Nepal. It is an unintentional termination of a pregnancy before the fetus has reached the seventh month of gestation [2,3]. It is a cultural taboo that receives little attention in the literature [4]. It is estimated that 8% to 15% of all clinically recognized pregnancies and 30% of all pregnancies result in spontaneous loss [2,5]. Statistical differences based on increasing maternal age pose an increasing risk of miscarriage among pregnant women [5]. Many pregnancies end in miscarriage before a women knows about her pregnancy status [2,4,6]. Investigations are usually conducted among those pregnant women having recurrent pregnancies [6]. The level of perinatal and maternal mortality may have been significantly reduced as well as other poor pregnancy outcomes in reproductive women as a result of improvements in the quality of care given during pregnancy. Yet, such improvements haven't had as much of an impact on the high miscarriage rate, with between 20% to 30% of pregnancies ending in miscarriage [7].

A study by Quenby et.al, revealed that the couple's age—both very young and older female age (less than 20 years and more than 35 years) as well as older males (more than 40 years)— may be associated with miscarriages [8]. Additionally, among pregnant women in reproductive life, very high or very low Body Mass Index (BMI), black race, prior miscarriages, smoking habits, alcohol consumption, stress, working night shifts, air pollution, and pesticide exposure were associated with miscarriages. Miscarriage involves both modifiable and non-modifiable risk factors [8]. According to a recent systematic review and meta-analysis of research undertaken in 26 different countries [2,9], active smoking along with obesity, are risk factors for miscarriage in Nepal [2]; caffeine use is also related to miscarriage and the presence of non-modifiable risk factors such as maternal age, chromosomal abnormalities, and aberrant uterine architecture are also associated factors for miscarriage. Fetal viability is significantly impacted by modifiable behavioral risk factors as well [2,10]. For instance, alcohol consumption during pregnancy was discovered to be associated with miscarriage [11,12] and smoking during pregnancy was associated with a slightly higher hazard ratio for miscarriage (1.18, 95% CI [0.96, 1.44]) [3,13]. In addition, drinking coffee while pregnant, heavy lifting, mental instability, health problems, and a history of abortion are all significant risk factors for miscarriages [14]. Even while several global and national initiative programs are working hard to reduce pregnancy-related risks and make motherhood safe, women still endure miscarriages. Various Nepal Demographic and Health Surveys (NDHSs) have revealed a shifting trend of miscarriage in Nepal [2], but understanding the causes of miscarriage in Nepal is still crucial.

The underlying etiology of miscarriage is still poorly known, although numerous researchers have worked to uncover possible risk factors [7,15]. There is less research that looks at the factors associated with miscarriage in low and middle-income countries (LMICs) like Nepal, as the majority of the information comes from high income nations. NDHS is the only source that provides data on miscarriage. However, factors associated with miscarriage are yet to be explored and analyzed. Therefore, a detailed examination of the root causes of miscarriage in Nepal is required. The study aims to examine the factors associated with miscarriage in Nepal by using the pooled data from the NDHS 2001, 2006, 2011, and 2016 to explore the various demographic, socio-economic, and maternal characteristics of miscarriage in the past 15 years.

The findings from this study would be helpful from the perspective of healthcare system planning to help government and non-governmental organizations modify current health policy and practices that focuses on miscarriage. This could be a significant step toward improving reproductive health in Nepal and achieving Sustainable Development Goal (SDG) Goal-3 to ensure healthy lives and promote well-being for all at all ages.

## Methods

### Data sources and sample composition

The datasets for the study were from NDHS 2001[16] 2006 [17], 2011 [18], and 2016 [19]. The NDHS is a nationally representative household survey using multistage cluster sampling designs, stratified by geographical regions, and urban and rural areas. All four surveys sampling methods were similar and routinely collected data to estimate socio-demographic, maternal and child health and mortality, fertility, HIV/AIDS, family planning, nutrition, and so on conducted every five years by the Ministry of Health of Nepal. The NDHS uses standardized techniques that involve standard questionnaires, manuals, and field procedures to collect data that is comparable across nations. Detailed standardized survey methodology and sampling methods are used in gathering the data [16–19]. The NDHS used three different types of surveys, each with information unique to the household, women, and men. The pre-tested, translated questionnaires were used to gather data on a variety of demographic and health variables, including a women's reproductive health outcome such as miscarriage, in three primary languages: Maithili, Bhojpuri, and Nepali.

When four of the NDHS surveys were pooled together there were a total of 45,055 women of reproductive age between 15 and 49 years, with an average response rate of nearly 97% [16–19]. The selection process for the sample from the NDHS in 2001, 2006, 2011, and 2016 is presented in Fig 1. Women of reproductive age were asked to record all pregnancies that resulted in both live and non-live births. Information on the duration of the pregnancy and the reason for termination was obtained for pregnancies that resulted in non-live births to determine if the pregnancy ended in a miscarriage or an induced abortion. A total weighted sample of 26,376 was obtained by limiting the analysis to pregnancies that terminated within the five years before the survey. This limitation was designed to reduce the mothers' recall bias, enhancing the study's internal and external validity.

### Outcome variable

Miscarriage is an outcome variable that refers to the spontaneous termination of a pregnancy before the fetus reaches the gestational age of seven months [1]. When a pregnancy ends before 7 months gestation, a miscarriage is classified as 1; 0 otherwise.

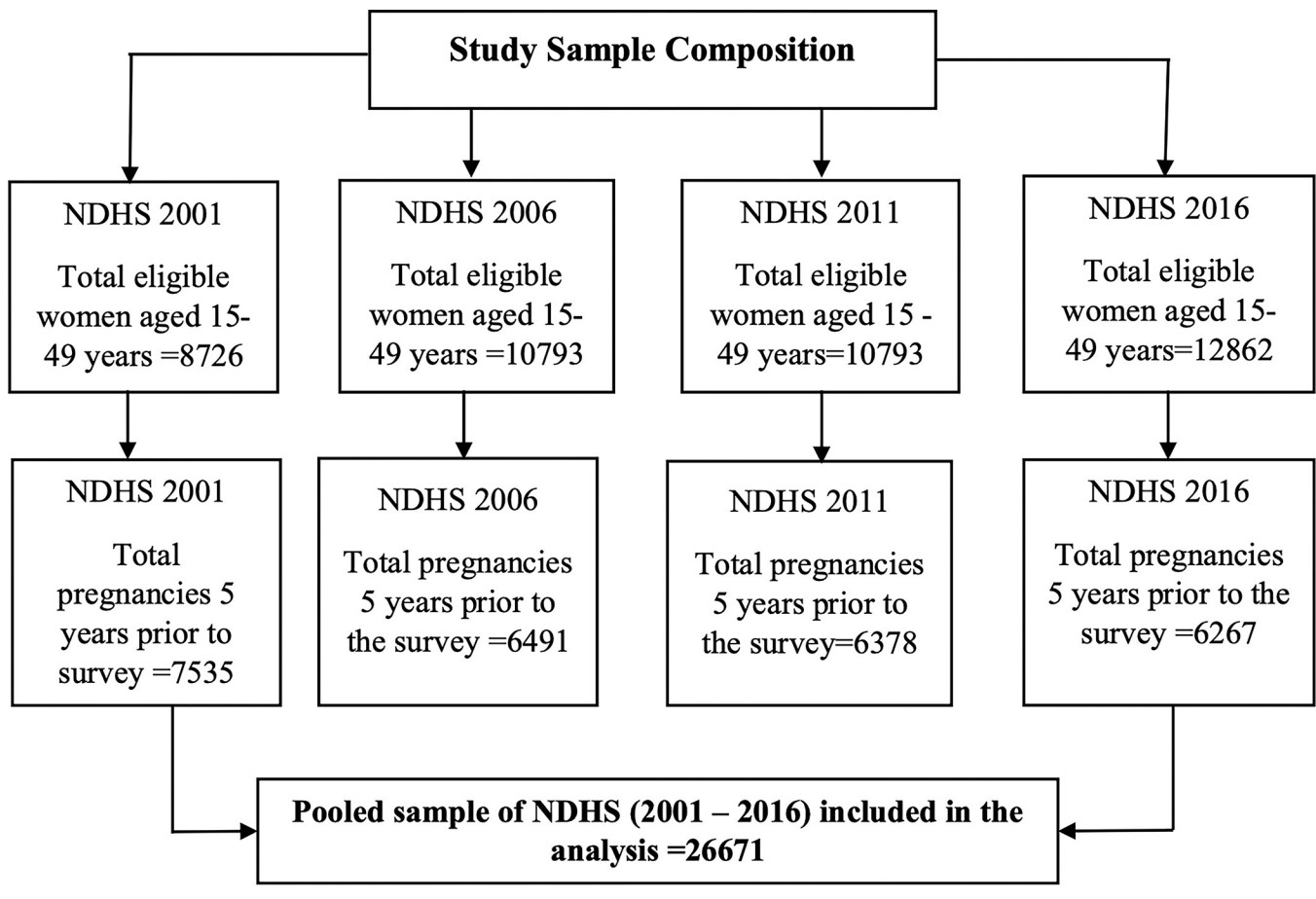

**Fig 1. Flow chart for selection of a sample from NDHS 2001, 2006, 2011, and 2016.**

### Study factors

We identified14 possible study variables classified into 3 different categories based on the availability of data from all four NDHS (2001–2016). Survey years 2001, 2006, 2011, and 2016 were the time-dependent confounding variables. The variables at the community level were place of residence (urban and rural), province (Koshi, Madhesh, Bagmati, Gandaki, Lumbini, Karnali, Sudurpaschim), and ecological zones (Mountain, Hill, and Terai). Socio-economic variables such as religion (Buddhist, Hindu and other religion), caste/ethnicity (Brahmin/Chhetri, Dalit, Janajati, Madhesi and others including Muslims), wealth index (Poorest, Poorer, Middle, Richer, Richest), women's educational status and their partner's educational level (No education, Primary, Secondary, Higher Secondary), their role as the head of the family (male headed family and female headed family), and their mother's employment (Not Working, Agriculture, Non Agriculture). Use of contraception, maternal smoking habit, and maternal age are exposure variables used to examine the proximate determinant for the outcome variables. They are grouped as maternal factors in the conceptual framework (Fig 2).

### Statistical analysis

The study was based on the deductive technique, and the analysis was carried out using a quantitative approach. The analytical frameworks for miscarriage have been conceptualized

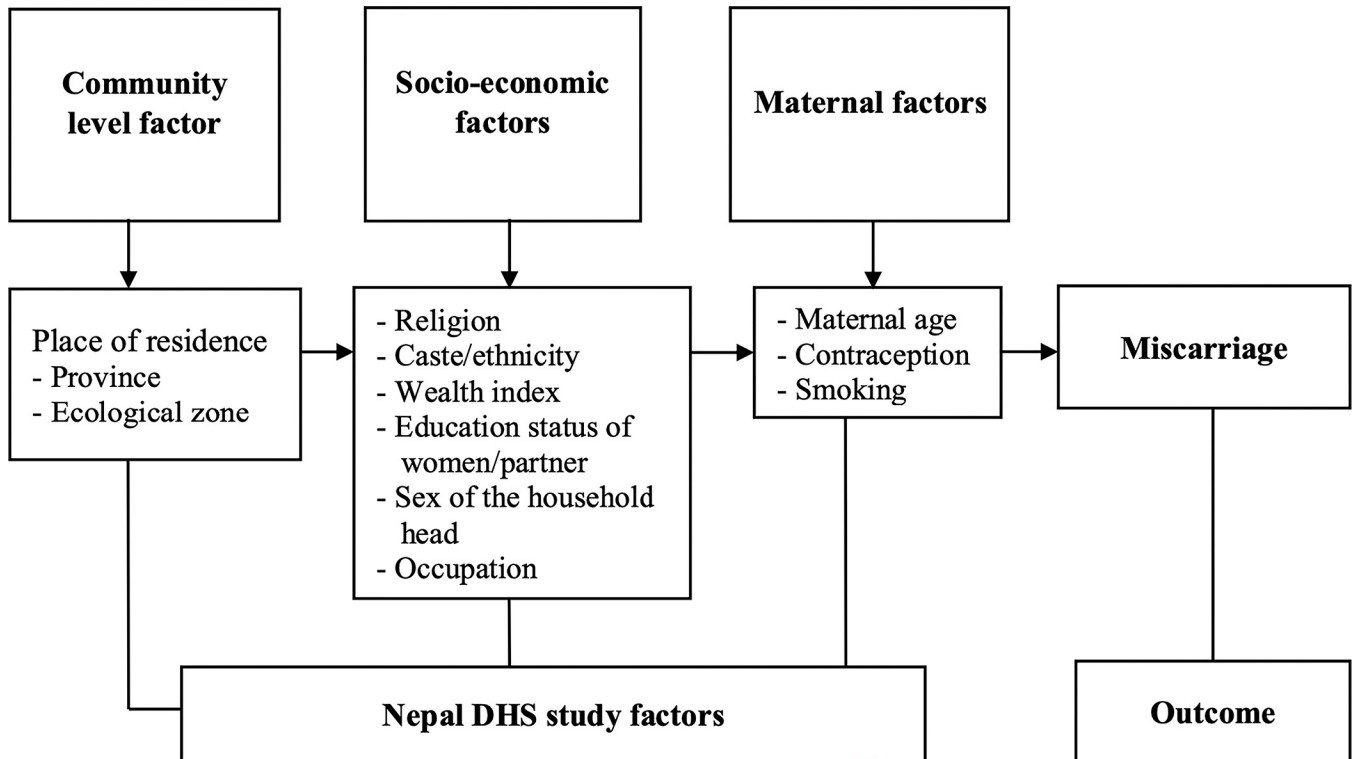

**Fig 2. Framework for factors affecting miscarriage in Nepal, adopted from Mosely & Chen's analytical framework for the study of child survival in developing countries.**

with multiple factors based on Mosley and Chen's analytical framework for child survival in developing countries [20] (Fig 2).

Multilevel logistic regression analysis that adjusted for the cluster and survey weights was used to identify the factors associated with miscarriage in Nepal, taking cluster and survey weights being taken into consideration based on Fig 2 [20,21]. Frequency tabulation was used to characterize the features of the study population. The prevalence with a 95% Confidence Interval (CI) of miscarriage were assessed for all study variables. The chi-square test was used to measure the association between the factor variables and miscarriage during the last 5 years and adjusted for the survey design that includes cluster and urban/rural stratification. Multi-collinearity amongst the predictor variables was checked using the Variance Inflation Factor (VIF). The mean value of VIF<10 was the cut-off point [22]. The statistical significance was considered at p-value < 0.05 and 95% confidence intervals (CIs).

Multivariable analysis was conducted by using a three-stage multilevel model (Fig 1) similar to those described to account for the complex hierarchical interrelationships between each blocks of determinants [21,23]. As part of hierarchical technique, we first analyzed variables from the community level block (Place of residence, Province and Ecological Zone) along with the survey year to establish a baseline multivariate model (Model I), Socio-economic Variables (Religion, Caste/Ethnicity, Wealth Index, Education Status of respondent, Education status of partner, Sex of the household head, Maternal occupation) were then fitted into Model I (Model II). In the final model (Model III), the exposure variable within maternal blocks (con-traception, maternal smoking habit and maternal age) was analyzed (Model II.).

The factors with p-values 0.05 in each step were kept. In order to avoid any statistical bias, we validated our findings by: (1) backward-eliminating potential risk factors with a p-value of

less than 0.20 from the univariate analysis; (2) testing the backward-elimination method by including all of the variables (all potential risk factors); and (3) testing and reporting collinearity. The odds ratios with 95% CI were performed to assess the adjusted risk of the independent variables, and those with p<0.05 were kept in the final model. The goodness of fit of the model was assessed by using Hosmer-Lemshow test. All analyses were performed using SPSS 26 version.

## Ethical considerations

NDHS received ethical approval from the Ethical Review Board of the Nepal Health Research Council and the ICF Institutional Review Board. The DHS website offers public access to the NDHS datasets [24]. The NDHS 2001–2016 datasets were made available for download and use after the author requested permission from MEASURE DHS/ICF International, Rockville, Maryland, USA. All research participants were read a pre-structured consent statement, and the interviewer verbally obtained their informed consent (assent on behalf of minors) was recorded by the interviewer.

## Results

Overall, 6.5% (1,715) (95% CI [6.2, 6.8]) of the pregnancies resulted in miscarriage during the last 15 years, based on the survey year 2001–2016. Different maternal age groups observed age-specific miscarriage and discovered a greater occurrence among women aged 45 to 49 years having a 10.7% prevalence (95% CI [7.07, 14.21]) followed by a roughly lower frequency of 10.3% among mothers between the ages of 40 to 44 (95% CI [8.30, 12.20]). Mothers aged 15 to 19 years also had a similar prevalence of 10.1% (95% CI [8.70, 11.46]) and a lower prevalence among age group 25–29, 5.8% (95% CI [5.26, 6.26]) whereas about 6.9% (95% CI [5.89, 8.00]) prevalence among age group 35–39, 6.0% prevalence among 30 to 34 age group and 6.1% (95% CI [5.54, 6.57]) prevalence of miscarriage observed among women from 20 to 24 years age groups (Fig 3).

## The basic profile of pregnant women with a proportion of miscarriage in Nepal in the last 15 years

NDHS data from 2001, 2006, 2011, and 2016 were combined to create a total of 26,376 pregnancies. Table 1 shows the weighted study population, number of miscarriage, miscarriage prevalence and 95% CI of study variables in Nepal. Over the fifteen years, the prevalence of miscarriage among pregnant women considerably rose from 4.9% (95% CI [4.3, 5.3]) in 2001 to 9.1% (95% CI [8.3, 9.8]) in 2016. The prevalence of miscarriage varied according to the age of the mother; which was highest among women older than 40 years (10.3%), followed by mothers younger than 20 years (10.1%), and lower among mothers aged 35 to 39 years (6.9%) and 20 to 34 years (5.9%) respectively. The prevalence of miscarriage was significantly higher for those living in urban areas (8.7%), Karnali Province (10.4%), and the hills (6.7%). Similarly, high prevalence was observed among those following the Hindu religion (6.6%), who belong to Brahmin/Chhetri (7.7%) and are richest in terms of wealth index (7.6%). Contraceptive non-users were also found to have a higher prevalence of miscarriage (7.5%).

## Factors associated with miscarriage in Nepal

Table 2 shows the unadjusted model, and three stages of multilevel models—Model I, Model II and Model III. Model III is resulting in a parsimonious model from Model I and Model-II in the multivariate analysis after adjusting for potential explanatory variables. The adjustment of

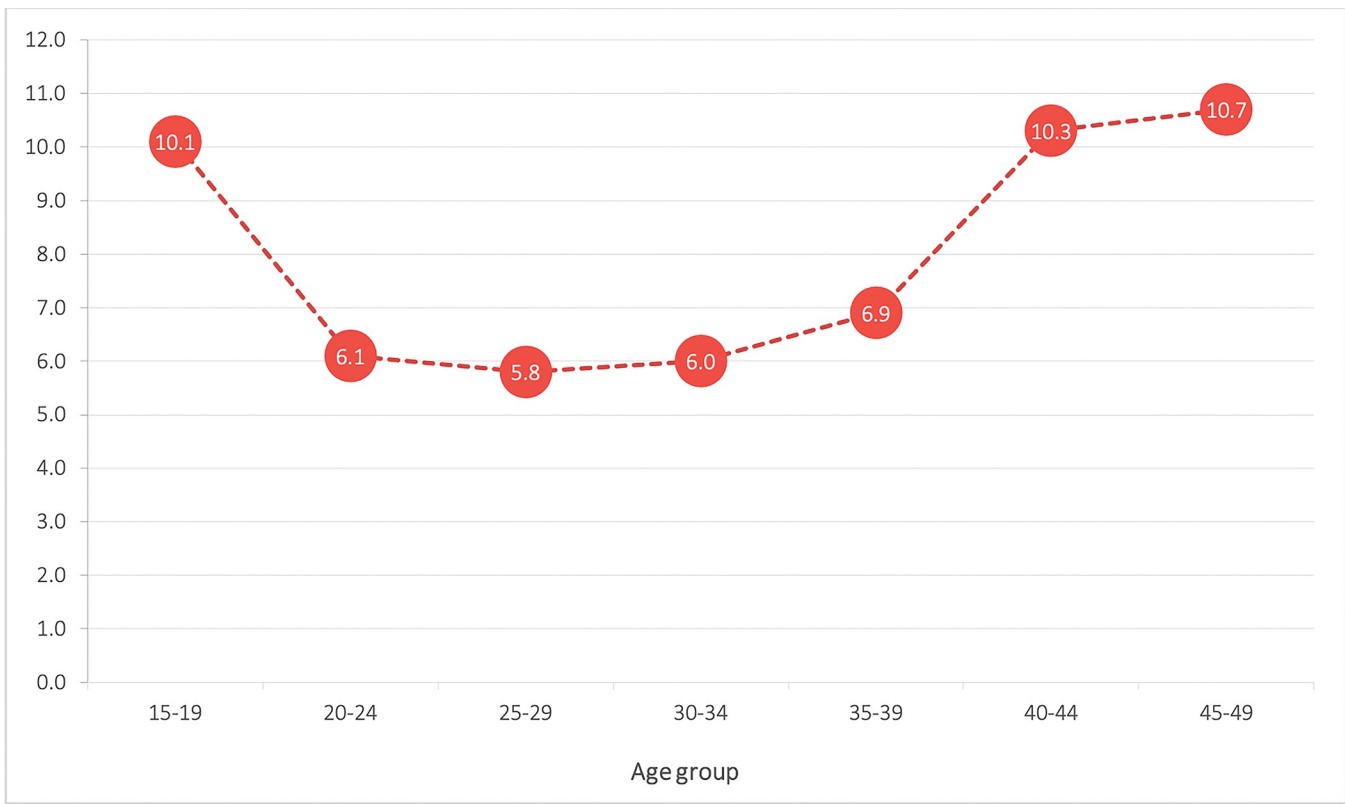

**Fig 3. Trends of miscarriage among women of reproductive age group (15 to 49) years based on maternal age.**

the variables is to identify the different factors of miscarriage in Nepal. This is based on the hierarchical approach shown in Fig 2. Most of the independent variables revealed associations in the unadjusted model (Table 2). Model adjusted for years of survey, place of residence, province, Ecological Zone, Religion, Caste/Ethnicity, Education status of women, partner's education, head of household head, maternal occupation, Contraception, maternal smoking habit and maternal age. Fixing multicollinearity by examining variance inflation factors with a cut-off value of <10. Some variables are manually removed from subsequent analysis to achieve model fit ($p > 0.05$).The goodness of fit of the model was assessed by using the Hosmer-Lemshow Test which showed the model was fit with statistically insignificant as the value of the test was 0.448 for Model I, 0.364 for Model II and 0.144 for Model III (Table 2).

The level of miscarriage has been steadily rising over the past 15 years, from 2001 to 2016. The likelihood of rising miscarriages is by 19% during 2001–2006 (aOR = 1.19, 95% CI [1.02, 1.39]), 50% in 2011 (aOR = 1.50, 95% CI [1.27–1.76]), and the chances increased by 88% in 2016 (aOR = 1.88, 95% CI [1.58–2.24]) since NDHS 2001 (Table 2 and Model III).

There was no statistically significant difference in miscarriage based on place of residence. But women living in urban areas had an odd of 0.87-times lower risk of getting a miscarriage (aOR = 0.87, 95% CI [0.76, 1.10]) compared with women from rural areas (Table 2 and Model III) though it was statistically significant in the bivariate analysis (unadjusted model).

In the multivariate analysis, the chance of miscarriage was demonstrated to be negligible with regards to respondents from various provinces, despite the fact that provinces exhibit a significant association with miscarriage in bivariate analysis ($p < 0.05$), particularly in the case of Lumbini and Karnali Province (Table 2, unadjusted model and Model III).

**Table 1. Weighted study population, miscarriage prevalence and 95% CI of study variables in Nepal (2001–2016), (N = 26376).**

| Explanatory Variables | Unweighted Population | Weighted Population | |
|---|---|---|---|
| | N*(26671) | N (26376) | Miscarriage % (95%CI) |
| Survey Year | | | |
| 2001 | 7535 | 7572 | 4.9 (4.3,5.3) |
| 2006 | 6491 | 6227 | 5.6 (5.0,6.2) |
| 2011 | 6378 | 6356 | 6.9 (6.1,7.3) |
| 2016 | 6267 | 6222 | 9.1 (8.3,9.8 |
| **Community Level Factors** | | | |
| Place of residence | | | |
| Urban | 7324 | 5431 | 8.7 (7.9–9.4) |
| Rural | 19347 | 20945 | 5.9 (5.6–6.2) |
| Province | | | |
| Koshi | 5155 | 4545 | 6.2 (5.5–6.9) |
| Madhes Province | 7430 | 8602 | 5.9 (5.3–6.3) |
| Bagmati Province | 5642 | 6914 | 6.7 (6.1–7.3) |
| Gandaki Province | 2837 | 2434 | 6.2 (5.2–7.1) |
| Lumbini Province | 3722 | 2895 | 7.8 (6.8–8.7) |
| Karnali Province | 1006 | 434 | 10.4 (7.5–13.3) |
| Sudurpaschim Province | 879 | 552 | 8.2 (5.9–10.5) |
| Ecological zone | | | |
| Tarai | 12271 | 13651 | 6.3 (5.9–6.7) |
| Hill | 10555 | 10664 | 6.7 (6.2–7.1) |
| Mountain | 3845 | 2061 | 6.6 (5.5–7.6) |
| **Socio-economic variables.** | | | |
| Religion | | | |
| Buddhist | 1698 | 1770 | 4,7 (3.7–5.7) |
| Hindu | 22954 | 22290 | 6.6 (6.3–6.9) |
| Others | 2019 | 2316 | 6.5 (5.47.4) |
| Caste/Ethnicity | | | |
| Brahmin/Chhetri | 8909 | 7492 | 7.7 (7.1–8.3) |
| Dalit | 4381 | 4173 | 6.7 (5.9–7.4) |
| Janajati | 8296 | 8496 | 5.4 (4.9–5.8) |
| Madhesi and others including Muslims | 5085 | 6216 | 6.4 (5.8–7.0) |
| Wealth index | | | |
| Poorest | 7294 | 6298 | 5.7 (5.1–6.3) |
| Poorer | 5475 | 5597 | 6.3 (5.7–6.9) |
| Middle | 4976 | 5427 | 6.4 (5.7–7.0) |
| Richer | 4854 | 4943 | 6.8 (6.0–7.4) |
| Richest | 4072 | 4111 | 7.6 (6.8–8.4) |
| Education Status of Respondents | | | |
| No education | 14355 | 14216 | 5.6 (5.2–5.9) |
| Primary | 4778 | 4806 | 7.5 (6.7–8.2) |
| Secondary | 1033 | 5911 | 7.4 (6.6–8.0) |
| Higher Secondary | 1507 | 1442 | 8.7 (7.1–10.0) |
| Sex of household head | | | |
| Male | 20903 | 20737 | 6.5 (6.1–6.7) |
| Female | 5768 | 5639 | 6.7 (6.0–7.3) |
| Maternal Working Status | | | |

*(Continued)*

**Table 1.** (Continued)

| Explanatory Variables | Unweighted Population | | Weighted Population |
|---|---|---|---|
| Not Working | 6037 | 6708 | 7.1 (6.4–7.7) |
| Agriculture | 17729 | 16803 | 6.0 (5.6–6.3) |
| Non-Agriculture | 2905 | 2865 | 8.2 (7.2–9.2) |
| **Maternal Factors(exposure Variables)** | | | |
| Contraception | | | |
| Using contraceptives | 9840 | 9645 | 4.7 (4.2–5.1) |
| Not using | 16831 | 16730 | 7.5 (7.1–7.9) |
| Smoking | | | |
| Smoker | 4355 | 4004 | 6.3 (5.5–7.0) |
| Non-smoker | 22316 | 22372 | 6.5 (6.2–6.8) |
| Maternal Age | | | |
| <20 years | 1780 | 1833 | 10.1 (8.7–11.4) |
| 20–34 years | 21326 | 21082 | 5.9 (5.6–6.2) |
| 35–39 years | 2324 | 2239 | 6.9 (5.8–8.0) |
| >40 years | 1241 | 1222 | 10.3 (8.6–12.0) |

N* Unweighted Population, N Weighted Population, weighted miscarriage %, CI = Confident Interval.

In the bivariate analysis (Table 2, unadjusted model), religion was found to be associated with miscarriage, but not in the multivariate analysis (Table 2, Model III). However, in comparison to Buddhists, women who belonged to other religion had a 17% likelihood of miscarriage (aOR = 1.05, 95% CI [0.77, 1.43]) whereas mothers who were Hindu had a 21.0% risk (aOR = 1.21, 95% CI [0.95 1.55]) of getting miscarriage, though it was not significantly associated ($p > 0.05$) at 95% CI.

The likelihood of getting a miscarriage on the basis of caste and ethnicity were shown to be strongly significant in bivariate analysis ($p < 0.05$) with all different sub categories of castes and ethnicities (Table 2, Unadjusted model). In the multivariate analysis (Model III and Table 2) among women belonging to Dalit groups the odd of getting miscarriage was 0.87 times less likely (aOR = 0.87, 95% CI [0.74, 1.02]), and similarly Janajatis also have an odd of 0.74 times less likely to get miscarriage (aOR = 0.74, 95% CI [0.64, 0.85]) as compare to women belonging to Brahmin/Chhetri caste groups, and it was also significantly associated ($p < 0.05$).

Both analysis—bivariate (unadjusted model, Table 2) and multivariate Model III (Table 2) ($p < 0.05$)—showed a significant relationship between the respondents' education status and their chance of miscarriage. When compared to the reference group of women with no education, only respondents with a primary level of education had 1.25 times greater odds of a miscarriage (aOR = 1.25, 95% CI [1.09, 1.44]) (Model III and Table 2).

With regards to the wealth index, bivariate and multivariate analyses both showed significant relationships ($p < 0.05$ at 95% CI) (Table 2). Women who belonged to different wealth indices had a considerably higher risk of miscarriage. The odds of miscarriage was 1.20 times higher (aOR = 1.20, 95% CI [1.02, 1.40]) for women who were in the poorer index, similarly, 1.23 times higher for Middle index (aOR = 1.23, 95% CI [1.03–1.44]), richer index 1.30 times higher odds (aOR = 1.30, 95% CI [1.09, 1.55]) and women in the richest wealth index have odds of 1.50 times greater (aOR = 1.50, 95% CI [1.22, 1.85]) than the reference group of women belongs to poorest wealth index.

The relationship between smoking and miscarriage was significant in the multivariate analysis but not in the bivariate analysis ($p > 0.05$) (Table 2). Women who did not smoke reported

**Table 2. Factors associated with miscarriage in Nepal, NDHS 2001–2016.**

| Correlates | Unadjusted Model | | Model I | | Model II | | Model III | |
|---|---|---|---|---|---|---|---|---|
| | OR(95%CI) | P Value | aOR(95%CI) | P Value | aOR(95%CI) | P Value | aOR(95%CI) | P Value |
| **Survey Year** | | | | | | | | |
| 2001* | 1 | | 1 | | 1 | | 1 | |
| 2006 | 1.16(1.00–1.35) | 0.50 | 1.14(0.98–1.32) | 0.08 | 1.13(0.97–1.31) | 0.11 | 1.19(1.02–1.39) | 0.02 |
| 2011 | 1.41(1.22–1.63) | 0.00 | 1.44(1.24–1.68) | 0.00 | 1.40(1.19–1.65) | 0.00 | 1.50(1.27–1.76) | 0.00 |
| 2016 | 1.94(1.70–2.23) | 0.00 | 1.72(1.47–2.02) | 0.00 | 1.72(1.45–2.04) | 0.00 | 1.88(1.58–2.24) | 0.00 |
| **Community level factors** | | | | | | | | |
| Rural* | 1 | | | | | | 1 | |
| Urban | 1.50(1.35–1.68) | 0.00 | 0.82(0.72–0.94) | 0.00 | 0.89(0.77–1.03) | 0.12 | 0.87(0.76–1.10) | 0.75 |
| **Province** | | | | | | | | |
| province 1* | 1 | | | | | | 1 | |
| Madhes Province | 0.92(0.79–1.07) | .0.33 | 0.89(0.76–1.03) | 0.13 | 0.88(0.75–1.03) | 0.12 | 0.87(0.74–1.02) | 0.95 |
| Bagmati Province | 1.08(0.93–1.26) | 0.28 | 1.00(0.85–1.18) | 0.95 | 0.97(0.82–1.14) | 0.72 | 0.96(0.81–1.13) | 0.67 |
| Gandaki Province | '0.99(0.81–1.22) | .0.97 | 1.03(0.84–1.26) | 0.75 | 0.99(0.80–1.22) | 0.97 | 0.96(0.78–1.18) | 0.71 |
| Lumbini Province | 1.27(1.06–1.52) | 0.00 | 1.16(0.96–1.39) | 0.11 | 1.12(0.93–1.35) | 0.21 | 1.10(0.91–1.33) | 0.30 |
| Karnali Province | 1.76(1.26–2.45) | 0.00 | 1.21(0.86–1.70) | 0.26 | 1.15(0.81–1.64) | 0.41 | 1.19(0.84–1.70) | 0.31 |
| Sudurpaschim Province | 1.34(0.97–1.86) | 0.07 | 0.90(0.64–1.27) | 0.57 | 0.84(0.59–1.19) | 0.33 | 0.88(0.62–1.24) | 0.46 |
| **Ecological zone** | | | | | | | | |
| Tarai* | 1 | | | | | | | |
| Hill | 1.06(0.96–1.18) | 0.21 | | | | | | |
| Mountain | 1.04(0.86–1.26) | 0.62 | | | | | | |
| **Socioeconomic Factors** | | | | | | | | |
| **Religion** | | | | | | | | |
| Buddhist* | 1 | | | | 1 | | 1 | |
| Hindu | 1.42(1.13–1.78) | 0.00 | | | 1.11(0.87–1.42) | 0.36 | 1.21(0.95–1.55) | 0.11 |
| Others | 1.38(1.05–1.82) | 0.02 | | | 1.86(0.86–1.56) | 0.31 | 1.17(0.86–1.58) | 0.29 |
| **Caste/Ethnicity** | | | | | | | | |
| Brahmin/Chhetri* | 1 | | | | 1 | | 1 | |
| Dalit | 0.85(0.73–0.99) | 0.04 | | | 0.92(0.78–1.08) | 0.34 | 0.87(0.74–1.02) | 0.10 |
| Janajati | 0.68(0.60–0.77) | 0.00 | | | 0.73(0.64–0.85) | 0.00 | 0.74(0.64–0.85) | 0.00 |
| Madhesi and others including Muslims | 0.82(0.72–0.93) | 0.00 | | | 0.84(0.71–0.98) | 0.34 | 0.82(0.69–0.96) | 0.16 |
| **Educational level of respondent** | | | | | | | | |
| No Education | 1 | | | | 1 | | 1 | |
| Primary Level Education | 1.36(1.19–1.54) | 0.00 | | | 1.20(1.05–1.38) | 0.00 | 1.25(1.09–1.44) | 0.00 |
| Secondary | 1.33(1.18–1.51) | 0.00 | | | 1.00(0.86–1.17) | 0.92 | 1.07(0.92–1.24) | 0.37 |
| Higher Secondary | 1.59(1.30–1.94) | 0.00 | | | 0.84(0.65–1.09) | 0.20 | 1.06(0.83–1.35) | 0.59 |
| **Education status of partner/husband** | | | | | | | | |
| No education* | 1 | | | | 1 | | | |
| primary | 1.10(0.95–1.27) | 0.16 | | | 0.99(0.85–1.15) | 0.93 | | |
| Secondary | 1.18(1.04–1.35) | 0.00 | | | 0.97(0.83–1.13) | 0.71 | | |
| Higher Secondary | 1.67(1.41–1.97) | 0.00 | | | 1.23(0.99–1.52) | 0.61 | | |
| **Wealth Index** | | | | | | | | |
| Poorest* | 1 | | | | 1 | | 1 | |
| Poorer | 1.11(0.95–1.29) | 0.17 | | | 1.12(0.96–1.31) | 0.14 | 1.20(1.02–1.40) | 0.02 |
| Middle | 1.12(0.97–1.31) | 0.11 | | | 1.13(0.96–1.33) | 0.12 | 1.23(1.03–1.44) | 0.01 |
| Richer | 1.19(1.02–1.39) | 0.02 | | | 1.14(0.96–1.36) | 0.12 | 1.30(1.09–1.55) | 0.00 |
| Richest | 1.35(1.15–1.58) | 0.00 | | | 1.20(0.97–1.48) | 0.85 | 1.50(1.22–1.85) | 0.00 |

*(Continued)*

**Table 2.** (Continued)

| Correlates | Unadjusted Model | | Model I | | Model II | | Model III | |
|---|---|---|---|---|---|---|---|---|
| | OR(95%CI) | P Value | aOR(95%CI) | P Value | aOR(95%CI) | P Value | aOR(95%CI) | P Value |
| **Sex of the household head** | | | | | | | | |
| Male | 1 | | | | | | | |
| Female | 1.03(0.92–1.16) | 0.55 | | | | | | |
| **Maternal Factor** | | | | | | | | |
| **Maternal currently working status** | | | | | | | | |
| Not Working | **1** | | | | **1** | | **1** | |
| Agriculture | 0.82(0.74–0.92) | 0.00 | | | 0.99(0.87–1.13) | 0.96 | 1.01(0.89–1.16) | 0.78 |
| Non Agriculture | 1.17(1.00–1.38) | 0.51 | | | 1.13(0.95–1.34) | 0.14 | 1.23(1.03–1.46) | 0.01 |
| **Exposure Variables** | | | | | | | | |
| Contraception | | | | | | | | |
| Used* | 1 | | | | | | 1 | |
| Not used | 1.64(1.47–1.83) | 0.00 | | | | | 1.81(1.61–2.03) | 0.00 |
| **Smoker** | | | | | | | | |
| Smoker* | 1 | | | | | | 1 | |
| Non-smoker | 1.04(0.90–1.194) | 0.57 | | | | | 0.81(0.69–0.95) | 0.01 |
| **Maternal Age** | | | | | | | | |
| <20 years | 1.78(1.51–2.09) | 0.00 | | | | | 1.71(1.44–2.02) | 0.00 |
| 20–34 years* | 1 | | | | | | 1 | |
| 35–39 years | 1.18(0.99–1.40) | 0.05 | | | | | 1.30(1.09–1.56) | 0.00 |
| >40 years | 1.83(1.51–2.22) | 0.00 | | | | | 2.12(1.73–2.59) | 0.00 |

Model I (Model adjusted for years of survey, Place of residence, Province).

Model II (Model adjusted for Years of survey, Place of residence, province, religion, caste/ethnicity, education status of women/partners, wealth index, maternal occupation).

Model III (Model adjusted for years of survey, place of residence, province, religion, caste/ethnicity, education status of women, wealth index, maternal occupation, contraception, smoking habit, maternal age).

0.81 times lower risks of getting a miscarriage than those who smoked (aOR = 0.81, 95% CI [0.69, 0.95]) (Table 2 and Model III).

Women who were not using contraception had higher probabilities of miscarriage, in both bivariate and multivariate analyses (Table 2). Women who did not use any form of contraception had a 1.81 times greater risk of miscarriage than those who did (aOR = 1.81, 95% CI [1.61, 2.03]) (Table 2 and Model III).

In both bivariate and multivariate analyses, the likelihood of a miscarriage was greater among women under the age of 20 and mothers older than 40 years old, with a somewhat lower chance among women in the age range of 35 to 39 years compared to the reference group of women aged 20 to 34 years, which is statistically significant as well ($p<0.05$) (Table 2). The odds of getting a miscarriage was 2.12 times higher among women more than 40 years (aOR = 2.12, 95% CI [1.73, 2.59]); similarly women belonging to less than 20 years younger also have odds of 1.71 times higher (aOR = 1.71, 95% CI [1.44, 2.02]) whereas among women belonging to age groups 35 to 39 years, there was 1.30 times greater odds of getting miscarriage (aOR = 1.30, 95% CI [1.09, 1.56]) with reference groups 20 to 34 years women (Table 2 and Model III).

## Discussion

During the last 15 years, from NDHS 2001 to 2016, 6.7% of pregnancies ended in miscarriage. We identified seven factors associated with miscarriage in Nepal namely, maternal age,

contraception use, maternal smoking behavior, caste or ethnicity, women pursuing elementary education, maternal occupation, and wealth index. They have a considerable impact on the likelihood of miscarriage among women of reproductive age groups in Nepal.

From the NDHS 2001 to 2016, the trends of miscarriages nearly doubled in Nepal, this findings is consistent with the study on pregnancy outcomes among Indian women which showed that the number of miscarriages among Indian women increased between 2015 and 2021 [25]. But in Finland the annual incidence of miscarriage among women aged 15 to 49 years has decreased significantly between 1998 and 2016 [5], falling by only 1.8% in 2016 from 1998, which is little bit contrast to the findings of our study in Nepal [5]. However, a study in India showed that, there were 6.3% of continuing pregnancies that resulted in miscarriage in last 3 years from 2014 to 2017 [20], and in the Manitoba study, the yearly miscarriage incidence was 11.3%, or around 1 in 9 pregnant women [26]. Though the findings are contextual, the main essence is that the trend of miscarriage is increasing in developing countries like Nepal but decreasing slightly in developed countries like Finland. We observed a U-shaped miscarriage trend with maternal age (Fig 3), with the risk of miscarriage being highest for women older than 40 and younger than 20 years. The rate of miscarriage was highest among women aged 15 to 19 and then decreased as women aged after turning 20, and almost stagnant till 34 years, increasing again as women aged over 35. Our study's findings are consistent with a birth cohort study from China [27] as well that found a J-shaped relationship between maternal age and spontaneous abortion, with advanced maternal age (>30 years) being significantly associated with miscarriage. However, in contrast, one study from Sudan found that the risk of miscarriage among Sudanese women follows a distinct curve in relation to maternal age, with the curve showing a lower risk for women under 20 years and at 40 years [28].

The study has supported the risk of miscarriage based on maternal age that the probability of miscarriage is higher among younger mothers (15–19 years), and then the probability of miscarriage sharply increased in older mothers (30+ years old) of reproductive age. Significant variations exist between reproductive women's age-specific groupings. While the increased risk of miscarriage at advanced maternal age could be the result of age-related hormonal changes, the significant increase in miscarriage among young women could be a reflection of biological phenomenon or it could reflect the hidden social context as well as the effect of reproductive immaturity. Further, we are in line with the findings of a Danish study that found that the risk of miscarriage is less than 15% until the age of 34, but increases to 25% between the ages of 35 and 39, 51% between the ages of 40 and 44, and more than 90% for women who are 45 years or older [29]. Chromosomal abnormalities, cessation of the uterine capacity, and depletion of ovarian follicles are all reasons why hormone treatment can be helpful for a woman trying to conceive later in life [30]. As chromosomal abnormalities are the most common cause of first-trimester miscarriage and are discovered in 50% to 80% of pregnancy tissues specimens after spontaneous miscarriage, a correlation between increasing maternal age and a higher incidence of chromosomal abnormality has been established in prior studies [29]. About two third of these are trisomies, and the likelihood of trisomy increases with maternal age. This study's findings are in line with those of Ford and MacCormac [30], as maternal aging is a significant, immutable factor in aneuploidy. It is linked to an increased risk of a live birth trisomy, particularly Down syndrome, and to a sharp rise in trisomy conceptions, the majority of which end in miscarriage.

The chance of miscarriage is also considerably lower among Dalit and Janajati women in Nepal than it is among Brahmin and Chhetri women, and this finding is important since it indicates that further research is needed to determine why this is the case in Nepal. We identified that women in Nepal with the highest wealth indices have a greater odds of miscarriage than those with the lowest wealth indices. Our findings from this study is supported by one

study in Bihar, India which revealed that there was an association between intimate partner violence (IPV) and miscarriage. Women in the lower wealth quartile (Quartile 1) showed no associations between IPV and miscarriage, but women in the higher wealth quartile (Quartile 3) saw an association between IPV and miscarriage [31]. Although, we are not examining intimate partner violence in this study, there is a possibility that it could be a hidden cause for miscarriage among women in the richest wealth quintile since they have higher odds of miscarriage in Nepal. The results are in stark contrast to a study of Danish and Chinese women, which found that those with higher salaries had a reduced incidence of spontaneous abortion than those with lower incomes [32,33]. Further research on contextual barriers for miscarriage is required.

When compared to women who utilized contraception, we identify that the likelihood of miscarriage was greater in the group of women who did not use contraceptions. Previous studies, found that a pattern of declining miscarriage incidence with increasing years of oral contraceptive (OC) use [34], and our study also shows similar results. But only for women over the age of 30, there was a significant relationship between the length of OC usage and miscarriage. The preservation of ovarian follicles caused by OC was formerly thought to be the cause of the 15% decrease in miscarriage rates among long-term pill users aged 30 or older and the following decrease in spontaneous abortion [34]. However, the current study's findings on the association between using contraception and miscarriage include all respondents who used both hormonal and non-hormonal methods of contraception.

Non-smoker women reported less chances of having a miscarriage than the reference group of respondents who smoked. The reason for this might be that smoking during pregnancy may influence the growth retardation of the fetus and the chance of having a miscarriage is increased. Active smoking increases the chance of miscarriage [2,9]. Another study, also found that smoking while pregnant increased the chance of miscarriage considerably [3,13].

The study has multiple benefits since its findings can add to the body of knowledge already available about the causes of miscarriage in Nepal. The study used data from nationally representative household surveys that were population-based and had a high response rate (>90%). The data were merged together to create a large sample size of miscarriage that was reported within 5 years preceding survey. Finally, this study applied appropriate statistical adjustments to data obtained from 4 nationally representative surveys and was able to identify the significant factors associated with miscarriage in Nepal.

This study has some limitations as well. First, this study is based on secondary data, and due to its cross-sectional nature, this paper is unable to establish a causal relationship between variables and occurrence of miscarriage. Second, the information on miscarriage is from retrospective data based on self-report from mothers which could be a potential source of recall and misclassification bias. Third, this study was not able to include important confounders such as the use of caffeine and alcohol, and obesity which have been previously identified as important modifiable risk factors for miscarriage in Nepal. Finally, miscarriage and other pregnancy complications might share underlying causes, which could be biological conditions or unmeasured common risk factors, hence, care should be taken in interpreting and applying the findings of this study.

## Conclusion

Our analyses examined factors associated with miscarriage in Nepal using pooled population-based surveys for the years 2001 to 2016.Miscarriage has increased significantly in Nepal. The likelihood of an increasing trend is close to two times higher in the data in NDHS 2016 than in NDHS 2001. Our study show that miscarriages are associated with maternal age, use of

contraception, smoking, wealth index, caste, and ethnicity. Interventions aimed to improve use of contraceptives, avoiding smoking and pregnancy planning on the basis of maternal age, are required to prevent miscarriage. Also, women who follow the Brahmin ethinicity and those with the highest income index require greater attention when it comes to miscarriage prevention strategies in Nepal.

Obesity in mothers is a significant contributor to miscarriage, and other studies have already identified it as a contributing factor in Nepal. However, the relationship of miscarriages with intimate partner violence is an important area that needs to be studied.

It is necessary to conduct more research to determine why miscarriage rates are rising in Nepal despite the nation's numerous safe motherhood and child health initiatives, as well as why elite (those with the highest wealth index) have a greater risk of miscarriage, along with intimate partner violence among highest wealth index. Further research is also necessary on the perceptions of women who have gone through a miscarriage in Nepal to examine the psychological impact among women of reproductive women, as this study was only able to quantify of the rates of miscarriages and the factors that affect it.

## Acknowledgments

The authors are thankful to measure DHS ICF International, and Rockville, Maryland, USA for granting access to the datasets used in this analysis. The authors would also like to acknowledge Dr. Meeta S. Pradhan for her efforts in editing the language.

## Author Contributions

**Conceptualization:** Sharadha Hamal, Yogendra B. Gurung, Prabin Shrestha.

**Data curation:** Sharadha Hamal.

**Formal analysis:** Sharadha Hamal, Prabin Shrestha, Nanda Lal Sapkota.

**Investigation:** Sharadha Hamal.

**Methodology:** Sharadha Hamal, Yogendra B. Gurung, Prabin Shrestha.

**Project administration:** Sharadha Hamal.

**Resources:** Sharadha Hamal.

**Software:** Sharadha Hamal.

**Supervision:** Sharadha Hamal.

**Validation:** Sharadha Hamal.

**Visualization:** Sharadha Hamal.

**Writing – original draft:** Sharadha Hamal, Yogendra B. Gurung, Bidhya Shrestha, Prabin Shrestha, Nanda Lal Sapkota, Vijaya Laxmi Shrestha.

**Writing – review & editing:** Sharadha Hamal, Yogendra B. Gurung, Bidhya Shrestha, Prabin Shrestha, Nanda Lal Sapkota, Vijaya Laxmi Shrestha.

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
