## [Decision Letter · Decision Letter 0]

11 Jul 2023

PONE-D-23-17092Factors Affecting Miscarriage in Nepal: Evidence from Nepal Demographic and Health Surveys, 2001-2016PLOS ONE

Dear Dr. Hamal

Thank you for submitting your manuscript to PLOS ONE. After careful consideration, we feel that it has merit but does not fully meet PLOS ONE’s publication criteria as it currently stands. Therefore, we invite you to submit a revised version of the manuscript that addresses the points raised during the review process.

We look forward to receiving your revised manuscript.

Kind regards,

Pradip Chouhan

Academic Editor

PLOS ONE

Journal Requirements:

Reviewers' comments:

Reviewer's Responses to Questions

**Comments to the Author**

1. Is the manuscript technically sound, and do the data support the conclusions?

Reviewer #1: No

Reviewer #2: Yes

2. Has the statistical analysis been performed appropriately and rigorously? 

Reviewer #1: No

Reviewer #2: Yes

3. Have the authors made all data underlying the findings in their manuscript fully available?

Reviewer #1: Yes

Reviewer #2: Yes

4. Is the manuscript presented in an intelligible fashion and written in standard English?

Reviewer #1: No

Reviewer #2: Yes

5. Review Comments to the Author

Reviewer #1: Thank you for providing me with the opportunity to review the manuscript titled "Factors Affecting Miscarriage in Nepal: Evidence from Nepal Demographic and Health Surveys, 2001-2016." I have carefully evaluated the manuscript and would like to acknowledge its contribution to the field. This is an interesting study which attempts to identify the factors associated with miscarriage in Nepal using data from four consecutive rounds of Nepal DHS.

However, I would also like to highlight several limitations that should be considered.

1. To begin with, I have an objection with the title of this study. The term "factors affecting" may not be the most suitable for a study that is not experimental in nature. When describing the relationship between variables in an observational study, it is more appropriate to use terms such as "associated with" or "associated factors" instead of "affecting."

By replacing "factors affecting" with "factors associated with," the revised title accurately reflects the nature of the study as an observational analysis of the relationships between variables rather than a direct causal investigation.

2. In line 84-88, you mentioned "The study found that the couple's age —" Could you please clarify which study you are referring to?

3. I would suggest that the authors use caution when using terms such as "have a role" or "were linked to," as they imply a causal relationship that can only be established through experimental studies.

4. In line 90, the authors have already stated that active smoking increases the chance of miscarriage. Therefore, there is no need to repeat this information in line 97.

5. Could you please provide further clarification on your statement in line 105-106: "Less conclusive and reliable data do exist, particularly in low- and middle-income nations like Nepal"? What specifically do you mean by "less conclusive" and "reliable data" in this context, as Nepal DHS is already there?

6. Based on the current conceptual framework of your study, which includes exposure variables (health service variables) and confounding variables, it seems that structural equation modeling (SEM) could be a suitable statistical analysis technique to address the complex relationships involved. SEM allows for the examination of both direct and indirect associations among variables, which is beneficial when dealing with multiple predictors and outcome variables.

Moreover, SEM can accommodate latent variables, which are constructs inferred from multiple observed indicators. This feature of SEM can be particularly valuable if your conceptual framework includes latent constructs that need to be considered.

Considering the complexity of your study's conceptual framework and the potential benefits of SEM, I would like to inquire if you have the necessary expertise and resources to perform a SEM analysis. If so, using SEM could provide a more comprehensive representation of your current conceptual framework and allow for a thorough investigation of the direct associations between the health service variables (exposure variables) and the outcome variable while considering the influence of confounding variables.

7. I would like to suggest considering the use of either multilevel modeling (MLM) or structural equation modeling (SEM) based on the hierarchical and nested structure of your data, as observed in the conceptual framework (Figure 2) and the design of the Demographic and Health Surveys (DHS).

MLM allows for the analysis of data with a hierarchical structure, such as individual-level, household-level, and community-level variables. It takes into account the dependencies and correlations within the nested data structure, providing a robust approach to analyzing such data. Given the presence of individual and community-level variables in your study, MLM could help account for the potential clustering effects and assess the influence of both individual and contextual factors on the outcome variable.

Alternatively, SEM could also be a suitable method for your study, considering its ability to handle complex relationships and pathways among multiple variables. SEM can accommodate latent variables and provide insights into direct and indirect associations. If there are latent constructs in your conceptual framework or if you aim to examine the direct associations between health service variables and the outcome variable while considering confounding variables, SEM could be a valuable approach.

I recommend selecting either MLM or SEM based on your convenience, expertise, and the specific goals of your study. Both methods offer unique advantages, and choosing the most suitable method will enhance the rigor and validity of your analysis.

8. Based on the authors' construction of separate models in lines 174-181, demonstrating an understanding of the hierarchy of variables, I recommend performing a multilevel modeling (MLM) analysis. MLM can effectively capture the nested structure of the data, allowing for examination of individual-level and higher-level effects. This approach will provide a comprehensive understanding of the factors influencing miscarriage and enhance the validity of the findings.

9. In the manuscript, it is not explicitly stated whether the authors checked for multicollinearity among the independent variables. However, I suggest that the authors perform a check for multicollinearity by examining the Variance Inflation Factors (VIFs) of the independent variables.

10. In ethical consideration section, authors should state that the access to the DHS datasets can be obtained through the DHS program's official website (https://dhsprogram.com/) by following the necessary protocols and permissions outlined by the program.

By providing this information, readers can be directed to the official DHS website where they can find further details on data access and the procedures to obtain the dataset for their own research purposes.

.

11. In Table 1, it is not explicitly stated whether the percentages are weighted. To ensure clarity and transparency in the reporting of results, I suggest including a footnote in Table 1 to indicate whether the percentages are weighted or unweighted.

Additionally, it would be beneficial to mention in the statistical analysis section of the study whether the percentages presented in the tables are weighted or unweighted.

12. The interpretation of odds ratios (ORs) in the paper (line 225-227) appears to be incorrect.

For example, if the adjusted odds ratio (AOR) is reported as 1.42, it indicates a 42% higher odds of a particular outcome occurring in a specific group compared to the reference group, rather than an absolute probability or a percentage increase.

Similarly, in your study, for the odds ratio of 2.06 for pregnant women over 40 years, it means that their odds of experiencing a miscarriage are 2.06 times higher than the reference group of pregnant women aged 20 to 34 years, rather than a 100% higher chance.

To ensure accurate interpretation, it is important to clarify in the paper that odds ratios represent relative changes in odds and not absolute probabilities or percentage increases. I recommend revising the results section accordingly to reflect the correct interpretation of odds ratios.

13. I would like to suggest reevaluating the placement of Table 2 in the manuscript. Since Table 2 primarily presents the results of unadjusted odds ratios, which are not intended to be interpreted or discussed extensively, it may be more appropriate to consider moving it to an appendix or removing it altogether.

14. Could you please explain the rationale behind including the variables in Table 2 (ecological zone, wealth index, sex of household head, and smoker) in further models, despite their lack of significant association with the outcome variable in the bivariate analysis?

In typical practice, bivariate analysis is often employed to identify variables that demonstrate a significant one-to-one relationship with the outcome variable. Variables that do not exhibit a significant association are typically excluded from further analysis. However, in your study, it appears that these variables have been included in subsequent models to compute adjusted odds ratios.

I recommend considering the removal of the statistically insignificant variables (ecological zone, wealth index, sex of household head, and smoker) from further models in your analysis. This approach helps mitigate potential issues related to multicollinearity and overfitting, ensuring that the final models accurately capture the predictors that have a meaningful impact on the outcome.

15. In lines 281-283 of the manuscript, it is mentioned that the authors compare their results with the findings of a study conducted in Finland regarding the annual incidence of miscarriage. I would like to inquire about the specific reason for exclusively comparing the results with Finland and not considering any other countries, particularly low- and middle-income countries (LMICs).

16. The discussion section of the manuscript requires significant improvement to enhance its clarity and logical flow. It is important to establish a coherent storyline that aligns with the study's objectives and effectively presents the key findings.

To address this, I recommend that the authors begin the discussion section by clearly stating the main objective of the study.

Subsequently, they should present the key findings in a structured and organized manner, discussing each finding in relation to the research objective and existing literature.

This will provide readers with a clear understanding of the significance and implications of the study's results. It is essential to provide a concise interpretation of the findings, highlighting their implications and contributions to the field.

Additionally, the authors should address any limitations or challenges encountered during the study and discuss their potential impact on the results and conclusions. Finally, the discussion section should be concluded by summarizing the main findings, their implications, and any recommendations for future research or practice.

By following these guidelines, the authors can significantly enhance the clarity, coherence, and overall quality of the discussion section, ensuring that it effectively communicates the study's objectives, findings, and implications to the readers.

17. In line 310, the authors use the term "Madhes Pradesh," while in other places it is referred to as "Madhes Province." This inconsistency in terminology can potentially confuse international readers.

To ensure clarity and avoid confusion, I recommend using a consistent and standardized term throughout the manuscript. Consider using the commonly recognized term "Madhes Province" to maintain consistency and make it easier for international readers to understand and follow the discussion.

18. I have noticed that the term "Dalit" is used in the manuscript to refer to a specific group of people. It is important to acknowledge that using such terminology can be value-laden and may carry certain connotations. I would like to suggest either using alternative terminology that is more neutral and inclusive or providing a clear justification for the use of the term "Dalit" in a footnote.

19. I have observed that in lines 313-316, the authors are restating their results within the discussion section of the manuscript. It is important to note that the discussion section should focus on the interpretation of the major findings and their implications, rather than reiterating the numerical results.

To enhance the quality and clarity of the discussion, I recommend the authors refrain from presenting the numerical results again in this section. Instead, they should emphasize the significance of the major findings and provide a comprehensive interpretation of their implications. Comparisons with relevant previous studies can be made to highlight similarities or differences in findings and to contribute to the existing body of knowledge.

By avoiding the repetition of numerical results and focusing on the interpretation and contextualization of the major findings, the discussion section will become more informative and insightful.

20. I have noticed that Reference 30 in the manuscript requires modification.

21. I recommend adding a footnote to each table in the manuscript to provide the full form of any abbreviations used.

22. The manuscript requires thorough proofreading and editing before it can be considered for publication. There are multiple grammatical errors throughout the paper that need to be addressed. These errors hinder the clarity and readability of the manuscript. Therefore, I recommend that the authors carefully review and revise the paper to ensure it meets the standards of scholarly writing.

Reviewer #2: Thank you for providing me with the opportunity to review the manuscript titled "Factors

Affecting Miscarriage in Nepal: Evidence from Nepal Demographic and Health Surveys,

2001-2016." (PONE-D-23-17092). After carefully reading the manuscript, I would like to

recommend the following changes for the improvement of the article.

1. The authors are advised to specify "The study found that the couple's age —" (Line

84). Proper citation is needed here.

2. The authors are asked not to repeat the same things. The smoking behaviour and

miscarriage are represented in lines 90 and 97.

3. The calculated percentages in Table 1 are not clearly mentioned whether these are

weighted or unweighted. Authors are advised to mention it.

4. Authors are asked to provide the rationale behind considering few variables like

ecological zone, wealth index, sex of household head etc. in this study.

5. The discussion section should focus on the explanation of the major findings and their

implications. So, the authors are advised to refrain from presenting the numerical

results.

6. PLOS authors have the option to publish the peer review history of their article (what does this mean?). If published, this will include your full peer review and any attached files.

Reviewer #1: No

Reviewer #2: No

---

## [Author Response · Author response to Decision Letter 0]

31 Aug 2023

The authors would like to thank the reviewers for their fair reviews and detailed list of constructive suggestions for the improvement of the manuscript. We greatly appreciate the opportunity to submit revisions of our paper based on the feedback received. We have revised the paper by providing more details based on comments received. Below, we have responded to each of the reviewers’ comments and the corresponding changes we made in the paper.

Reviewer 1 (R1) concerns:

1. To begin with, I have an obligation with the title of this study. The term "Factors affecting" may not be the most suitable for the study that is not experimental in nature. When describing the relationship between variables in an observational study. it is more appropriate to use terms such as "associated with" or "associated factors" instead of "affecting" by replacing "factors affecting" with "factors associated with" the revised the title accurately between variables rather than direct causal investigation.

Our Response: We revised the title by replacing "factors affecting" with "factors associated" in our title.

2. In line 84-88, you mentioned "The study found that the couple's age —" Could you please clarify which study you are referring to?

Our Responses: This is cited from research titled "Miscarriage Matters: The Epidemiological, Physical, Psychological, and Economic Costs of Early Pregnancy Loss," which was published in Lancet 397: 1658–67 in 2021. According to the study, a couple's age means females aged less than 20 years and more than 35 years and with a male (husband) older than 35 years, there is a worldwide risk of miscarriage. We have provided the reference of the study as well. 

3. I would suggest that the authors use caution when using terms such as "have a role" or "were linked to," as they imply a causal relationship that can only be established through experimental studies.

Our responses: we fixed the phrases "have a role" and "were link to" by substituting it with the phrase associated with.

4. In line 90, the authors have already stated that active smoking increases the chance of miscarriage. Therefore, there is no need to repeat this information in line 97.

Our Responses: We have fixed the duplication of information in lines 90 and 97.

5. Could you please provide further clarification on your statement in line 105-106: "Less conclusive and reliable data do exist, particularly in low- and middle-income nations like Nepal"? What specifically do you mean by "less conclusive" and "reliable data" in this context, as Nepal DHS is already there?

Our Responses: We do concur that Nepal DHS is present. However, the DHS lacks sufficient data on miscarriage. It has restrictions on a number of significant elements, such as certain biological ones, which may directly contribute to miscarriage. In Nepal, there have been little studies on miscarriage.

6. Based on the current conceptual framework of your study, which includes exposure variables (health service variables) and confounding variables, it seems that structural equation modeling (SEM) could be a suitable statistical analysis technique to address the complex relationships involved. SEM allows for the examination of both direct and indirect associations among variables, which is beneficial when dealing with multiple predictors and outcome variables.

Moreover, SEM can accommodate latent variables, which are constructs inferred from multiple observed indicators. This feature of SEM can be particularly valuable if your conceptual framework includes latent constructs that need to be considered.

Considering the complexity of your study's conceptual framework and the potential benefits of SEM, I would like to inquire if you have the necessary expertise and resources to perform a SEM analysis. If so, using SEM could provide a more comprehensive representation of your current conceptual framework and allow for a thorough investigation of the direct associations between the health service variables (exposure variables) and the outcome variable while considering the influence of confounding variables.

Our responses: We have conceptualized our conceptual framework again on the basis of Mosley and Chen's analytical framework as described by the study (Factors associated with perinatal mortality in Nepal: evidence from Nepal demographic and health survey 2001–2016). And carried out multilevel logistic regression analysis adjusted for cluster and survey weight used to identify the significant factor associated with miscarriage in Nepal.

7. I would like to suggest considering the use of either multilevel modeling (MLM) or structural equation modeling (SEM) based on the hierarchical and nested structure of your data, as observed in the conceptual framework (Figure 2) and the design of the Demographic and Health Surveys (DHS).

MLM allows for the analysis of data with a hierarchical structure, such as individual-level, household-level, and community-level variables. It takes into account the dependencies and correlations within the nested data structure, providing a robust approach to analyzing such data. Given the presence of individual and community-level variables in your study, MLM could help account for the potential clustering effects and assess the influence of both individual and contextual factors on the outcome variable.

Alternatively, SEM could also be a suitable method for your study, considering its ability to handle complex relationships and pathways among multiple variables. SEM can accommodate latent variables and provide insights into direct and indirect associations. If there are latent constructs in your conceptual framework or if you aim to examine the direct associations between health service variables and the outcome variable while considering confounding variables, SEM could be a valuable approach.

I recommend selecting either MLM or SEM based on your convenience, expertise, and the specific goals of your study. Both methods offer unique advantages, and choosing the most suitable method will enhance the rigor and validity of your analysis.

Our Responses: Please look at the response 8 for details. 

8. Based on the authors' construction of separate models in lines 174-181, demonstrating an understanding of the hierarchy of variables, I recommend performing a multilevel modeling (MLM) analysis. MLM can effectively capture the nested structure of the data, allowing for examination of individual-level and higher-level effects. This approach will provide a comprehensive understanding of the factors influencing miscarriage and enhance the validity of the findings.

Our Responses: On the basis of our expertise multilevel logistic regression analysis was used to identify the factors associated with miscarriage in Nepal, taking cluster and survey weights being taken into consideration based on Fig 2 in our revised manuscript. Multivariable analysis was conducted by using a three-stage multilevel model (Figure 2) similar to those described to account for the complex hierarchical interrelationships between each block of determinants [21, 23] in the reference list of the revised manuscript. As part of the hierarchical technique, we first analyzed variables from the community level block (Place of residence, Province, and ecological Zone) along with the survey year to establish a baseline multivariate model (model I), Socioeconomic Variables (Religion, Caste/Ethnicity, Wealth Index, Education Status of the respondent, Education status of the partner, sex of the household head, maternal occupation) were then fitted into model 1 (model 2). In the final model (model 3), the exposure variables within maternal blocks (conception, maternal smoking habit, and maternal age)) were analyzed with model 2. All the variables are weighted by women's individual sample weight V005 in the DHS data set. These are described in Table 1 and Table 2 of the revised manuscript.

9. In the manuscript, it is not explicitly stated whether the authors checked for multicollinearity among the independent variables. However, I suggest that the authors perform a check for multicollinearity by examining the Variance Inflation Factors (VIFs) of the independent variables.

Our Responses: We have examined the Variance Inflation Factors (VIF) and tested for multi-collinearity.

10. In ethical consideration section, authors should state that the access to the DHS datasets can be obtained through the DHS program's official website (https://dhsprogram.com/) by following the necessary protocols and permissions outlined by the program.

By providing this information, readers can be directed to the official DHS website where they can find further details on data access and the procedures to obtain the dataset for their own research purposes.

Our Responses: We have made corrections in response to your input.

11. In Table 1, it is not explicitly stated whether the percentages are weighted. To ensure clarity and transparency in the reporting of results, I suggest including a footnote in Table 1 to indicate whether the percentages are weighted or unweighted. Additionally, it would be beneficial to mention in the statistical analysis section of the study whether the percentages presented in the tables are weighted or unweighted.

Our responses: Prior drafts of the manuscripts had unweighted percentages, however, in the revised draft, we weighted the percentage and added a footnote accordingly. 

12. The interpretation of odds ratios (ORs) in the paper (line 225-227) appears to be incorrect.

For example, if the adjusted odds ratio (AOR) is reported as 1.42, it indicates a 42% higher odds of a particular outcome occurring in a specific group compared to the reference group, rather than an absolute probability or a percentage increase.

Similarly, in your study, for the odds ratio of 2.06 for pregnant women over 40 years, it means that their odds of experiencing a miscarriage are 2.06 times higher than the reference group of pregnant women aged 20 to 34 years, rather than a 100% higher chance.

To ensure accurate interpretation, it is important to clarify in the paper that odds ratios represent relative changes in odds and not absolute probabilities or percentage increases. I recommend revising the results section accordingly to reflect the correct interpretation of odds ratios.

Our Responses: We have made necessary correction in revised version of the manuscript.

13. I would like to suggest reevaluating the placement of Table 2 in the manuscript. Since Table 2 primarily presents the results of unadjusted odds ratios, which are not intended to be interpreted or discussed extensively, it may be more appropriate to consider moving it to an appendix or removing it altogether.

Our Response: Separate bivariate analysis tables were removed, however, final table 2 was included as an unadjusted model in the revised manuscript for comparison of analysis.

14. Could you please explain the rationale behind including the variables in Table 2 (ecological zone, wealth index, sex of household head, and smoker) in further models, despite their lack of significant association with the outcome variable in the bivariate analysis?

In typical practice, bivariate analysis is often employed to identify variables that demonstrate a significant one-to-one relationship with the outcome variable. Variables that do not exhibit a significant association are typically excluded from further analysis. However, in your study, it appears that these variables have been included in subsequent models to compute adjusted odds ratios. I recommend considering the removal of the statistically insignificant variables (ecological zone, wealth index, sex of household head, and smoker) from further models in your analysis. This approach helps mitigate potential issues related to multicollinearity and overfitting, ensuring that the final models accurately capture the predictors that have a meaningful impact on the outcome.

Our responses: We conducted reanalysis of the revised manuscript from the weighted sample. The factors with p-values of 0.05 in each step were kept. In order to avoid any statistical bias, we validated our findings by (1) backward-eliminating potential risk factors with a p-value of less than 0.20 from the univariable analysis; (2) testing the backward-elimination method by including all of the variables (all potential risk factors); and (3) testing and reporting collinearity. The odds ratios with 95% CI were performed to assess the adjusted risk of the independent variables, and those with p< 0.05 were kept in the final model. The goodness of fit of the model was assessed by using Hosmer- Lemshow test. We kept smoking in the final model even though bivariate analysis had a p>0.05 significance level because smoking would be a proximate determinant that is mentioned as a maternal factor in our conceptual framework and was already established as a factor for miscarriage by other studies. However, we wanted to check the validity of these variables in our study, so we checked for multicollinearity and discovered that there was no multicollinearity due to these variables. Therefore, we used these factors in our analysis.

15. In lines 281-283 of the manuscript, it is mentioned that the authors compare their results with the findings of a study conducted in Finland regarding the annual incidence of miscarriage. I would like to inquire about the specific reason for exclusively comparing the results with Finland and not considering any other countries, particularly low- and middle-income countries (LMICs).

Our responses: Along with the study from Finland, we have added one study from a developing nation. Through this, we can discuss and compare our study findings both from developed and developing nations. We have now incorporated a new study from low-and middle-income countries (LMICs) in the discussion.

16. The discussion section of the manuscript requires significant improvement to enhance its clarity and logical flow. It is important to establish a coherent storyline that aligns with the study's objectives and effectively presents the key findings. To address this, I recommend that the authors begin the discussion section by clearly stating the main objective of the study.

Subsequently, they should present the key findings in a structured and organized manner, discussing each finding in relation to the research objective and existing literature. This will provide readers with a clear understanding of the significance and implications of the study's results. It is essential to provide a concise interpretation of the findings, highlighting their implications and contributions to the field. Additionally, the authors should address any limitations or challenges encountered during the study and discuss their potential impact on the results and conclusions. Finally, the discussion section should be concluded by summarizing the main findings, their implications, and any recommendations for future research or practice. By following these guidelines, the authors can significantly enhance the clarity, coherence, and overall quality of the discussion section, ensuring that it effectively communicates the study's objectives, findings, and implications to the readers.

Our Responses: We have made an effort to tailor our discussion to your suggestions in the discussion area so that it would clearly convey the study's goals, findings, and implications to readers.

17. In line 310, the authors use the term "Madhes Pradesh," while in other places it is referred to as "Madhes Province." This inconsistency in terminology can potentially confuse international readers.

To ensure clarity and avoid confusion, I recommend using a consistent and standardized term throughout the manuscript. Consider using the commonly recognized term "Madhes Province" to maintain consistency and make it easier for international readers to understand and follow the discussion.

Our Response: we have made corrections by consistently and evenly employing words. Making "Madhesh Province" rather than "Madesh Pradesh" to minimize confusion.

18. I have noticed that the term "Dalit" is used in the manuscript to refer to a specific group o

---

## [Decision Letter · Decision Letter 1]

25 Oct 2023

PONE-D-23-17092R1Factors Associated with Miscarriage in Nepal: Evidence from Nepal Demographic and Health Surveys, 2001-2016PLOS ONE

Dear Dr. Sharadha Hamal,

Thank you for submitting your manuscript to PLOS ONE. After careful consideration, we feel that it has merit but does not fully meet PLOS ONE’s publication criteria as it currently stands. Therefore, we invite you to submit a revised version of the manuscript that addresses the points raised during the review process.

We look forward to receiving your revised manuscript.

Kind regards,

Pradip Chouhan

Academic Editor

PLOS ONE

Reviewers' comments:

Reviewer's Responses to Questions

**Comments to the Author**

1. If the authors have adequately addressed your comments raised in a previous round of review and you feel that this manuscript is now acceptable for publication, you may indicate that here to bypass the “Comments to the Author” section, enter your conflict of interest statement in the “Confidential to Editor” section, and submit your "Accept" recommendation.

Reviewer #3: All comments have been addressed

Reviewer #4: (No Response)

2. Is the manuscript technically sound, and do the data support the conclusions?

Reviewer #3: Yes

Reviewer #4: (No Response)

3. Has the statistical analysis been performed appropriately and rigorously? 

Reviewer #3: Yes

Reviewer #4: (No Response)

4. Have the authors made all data underlying the findings in their manuscript fully available?

Reviewer #3: Yes

Reviewer #4: (No Response)

5. Is the manuscript presented in an intelligible fashion and written in standard English?

Reviewer #3: No

Reviewer #4: (No Response)

6. Review Comments to the Author

Reviewer #3: This is a great piece of work in a topic where limited research are done, especially in resource poor settings like Nepal. This manuscript may benefit from some minor revisions as described below.

Abstract

Methods: write factors associated instead of significant factors

Results: please remove the first sentence to reflect more on your study title.

Conclusion: please be specific with your results when you draw the conclusion

Introduction: Please look at English grammar throughout.

Methods: Please combine exposure variables and confounding variables into one heading under study factors.

Analysis: remove ‘Method of analysis’ and replace with ‘Statistical analysis’

Figure/table headings: Please re-write your fig/table headings (for example: Table 2: Factors associated with miscarriage in Nepal, NDHS 2001-2016)

Please be consistent with the choice of word. For example, if your title is factors associated with miscarriage, please use them throughout rather than replacing it with word such as predictor.

Reviewer #4: The paper have been designed with the help of DHS 2001-2016, but the recent demographic data of Nepal (DHS-2022) is available and the analysis can be done with the help of recent published data which will have more applicability in terms of present date. Therefore, the latest data should be incorporated in the study. More factor variables (obesity status of women) can be used as explanatory variables against miscarriage.

The study shows that miscarriage rate of Nepal from 2001 to 2016 is increased by 4.2 percentage points. Proper explanation is very necessary regarding the gradual increasing of miscarriage rate from 2001 to 2016, in spite of having improved medical facilities day by day.

7. PLOS authors have the option to publish the peer review history of their article (what does this mean?). If published, this will include your full peer review and any attached files.

Reviewer #3: **Yes: **Pramesh Ghimire

Reviewer #4: No

---

## [Author Response · Author response to Decision Letter 1]

8 Dec 2023

The authors would like to thank the reviewers for their fair reviews and detailed list of constructive Suggestions. We greatly appreciate the opportunity to submit a major revision of our paper for your renewed consideration. We have given more detail based on comments we are receiving from you. Below, we respond to reviewers’ comments and point to the corresponding changes we made in the paper.

Reviewer (R3) concerns:

Reviewer #3: This is a great piece of work in a topic where limited research are done, especially in resource poor settings like Nepal. This manuscript may benefit from some minor revisions as described below.

Abstract

Methods: write factors associated instead of significant factors

Results: please remove the first sentence to reflect more on your study title.

Conclusion: please be specific with your results when you draw the conclusion

Introduction: Please look at English grammar throughout.

Methods: Please combine exposure variables and confounding variables into one heading under study factors.

Analysis: remove ‘Method of analysis’ and replace with ‘Statistical analysis’

Figure/table headings: Please re-write your fig/table headings (for example: Table 2: Factors associated with miscarriage in Nepal, NDHS 2001-2016) 

Please be consistent with the choice of word. For example, if your title is factors associated with miscarriage, please use them throughout rather than replacing it with word such as predictor.

Our responses

Abstract

Methods: We have changed factor associated instead of significant factors in methods section as per your suggestion.

Results: We removed first sentence from result section.

Conclusion: We revised our prior conclusion and were more specific with the findings of our study in new revised manuscript.

Introduction: We worked with a professional paper editor to revise our article and make grammatical and sentence structure corrections.

Methods: We made the changes you suggested put in one heading under study factors.

Analysis: We made correction and replacing methods of analysis in analysis section.

Figure/table headings: In response to your comments, we reworked our figure/table title to be shorter and more concise.

Please be consistent with the choice of word. For example, if your title is factors associated with miscarriage, please use them throughout rather than replacing it with word such as predictor.

We have tried to become consistent in our word choice throughout the article. We changed the word predictor to the factor associated with it to maintain consistency across the study.

Reviewer #4 :( concerns)

Reviewer #4: The paper have been designed with the help of DHS 2001-2016, but the recent demographic data of Nepal (DHS-2022) is available and the analysis can be done with the help of recent published data which will have more applicability in terms of present date. Therefore, the latest data should be incorporated in the study. More factor variables (obesity status of women) can be used as explanatory variables against miscarriage. The study shows that miscarriage rate of Nepal from 2001 to 2016 is increased by 4.2 percentage points. Proper explanation is very necessary regarding the gradual increasing of miscarriage rate from 2001 to 2016, in spite of having improved medical facilities day by day.

Our Responses:

The authors would like to thank you for your suggestions for the study. Of course, the most recent data DHS-2022 report is already accessible. The study used data from the NDHS 2001 to the NDHS 2016 since the study is based on the Millennium Development Goal era, which is from 2000 to 2015, the finding would assist to establish strategies for achieving Sustainable Development Goal (SDG) Goal-3 to ensure healthy lives and promote well-being for all at all ages. And also we have sufficient sample size around 26,376 which is sufficient enough to calculate power. It is like similar proportion of Miscarriage in DHS 2022 in Nepal is about 9.4% almost near to equal level of proportion with DHS 2016. The data was released after the COVID-19 pandemic, therefore it is possible that it was impacted by the COVID-19 pandemic. Thank you for your suggestion; I will include it in a future publication as I am preparing my next paper on miscarriage based on DHS 2022 data. The variable obesity status of women has previously been proven as a risk factor for miscarriage in Nepal by Pramesh et al on an association between obesity and miscarriage in Nepal based on data from 2001 to 2016, we were unable to include it in our study. The DHS data shows that miscarriage is increasing in Nepal and there are various reasons associated with it. Limited access to maternal health care services, Poor quality of care, malnutrition, poor maternal health are the factors contributing to increased miscarriage in Nepal, however this is still an area that needs to be explored.

---

## [Decision Letter · Decision Letter 2]

21 Feb 2024

PONE-D-23-17092R2Factors Associated with Miscarriage in Nepal: Evidence from Nepal Demographic and Health Surveys, 2001-2016PLOS ONE

Dear Dr. Hamal,

Thank you for submitting your manuscript to PLOS ONE. After careful consideration, we feel that it has merit but does not fully meet PLOS ONE’s publication criteria as it currently stands. Therefore, we invite you to submit a revised version of the manuscript that addresses the points raised during the review process.

We look forward to receiving your revised manuscript.

Kind regards,

Ganesh Dangal, MD, FICS, FRCOG

Academic Editor

PLOS ONE

Journal Requirements:

Reviewers' comments:

Reviewer's Responses to Questions

**Comments to the Author**

1. If the authors have adequately addressed your comments raised in a previous round of review and you feel that this manuscript is now acceptable for publication, you may indicate that here to bypass the “Comments to the Author” section, enter your conflict of interest statement in the “Confidential to Editor” section, and submit your "Accept" recommendation.

Reviewer #3: All comments have been addressed

2. Is the manuscript technically sound, and do the data support the conclusions?

Reviewer #3: Yes

3. Has the statistical analysis been performed appropriately and rigorously? 

Reviewer #3: Yes

4. Have the authors made all data underlying the findings in their manuscript fully available?

Reviewer #3: Yes

5. Is the manuscript presented in an intelligible fashion and written in standard English?

Reviewer #3: Yes

6. Review Comments to the Author

Reviewer #3: Thanks for addressing all the comments. The revised version seems much better shape. I am sure that this manuscript will have greater impact to reduce miscarriage in Nepal.

7. PLOS authors have the option to publish the peer review history of their article (what does this mean?). If published, this will include your full peer review and any attached files.

Reviewer #3: No

---

## [Author Response · Author response to Decision Letter 2]

25 Mar 2024

We have reviewed the list of references. There are no any references that have been retracted. One of the references that we cited earlier have been removed.

---

## [Editor Report · Decision Letter 3]

27 Mar 2024

Factors Associated with Miscarriage in Nepal: Evidence from Nepal Demographic and Health Surveys, 2001-2016

PONE-D-23-17092R3

Dear Dr. Hamal,

We’re pleased to inform you that your manuscript has been judged scientifically suitable for publication and will be formally accepted for publication once it meets all outstanding technical requirements.

Kind regards,

Ganesh Dangal, MD, FICS, FRCOG

Academic Editor

PLOS ONE